# Polymorphism and Mechanical Behavior in Hot-Pressed 3D-Printed Polyamide Composite: Effects of Pressure and Temperature

**DOI:** 10.3390/polym17070922

**Published:** 2025-03-28

**Authors:** John Barber, Patricia Revolinsky, Jimesh Bhagatji, Diego Pedrazzoli, Sergii Kravchenko, Oleksandr Kravchenko

**Affiliations:** 1Composites Modeling and Manufacturing Group, Department of Mechanical and Aerospace Engineering, Old Dominion University, Norfolk, VA 23529, USA; 2Johns-Manville Corp., Denver, CO 80202, USA; diego.pedrazzoli@jm.com; 3Department of Materials Engineering, The University of British-Columbia, Vancouver, BC V6T 1Z4, Canada; sergey.kravchenko@ubc.ca

**Keywords:** 3D printing, high-temperature compaction, polymorphism, crystallization

## Abstract

The aim of this work is to study the effect of high-temperature compaction (HTC) upon the polymorphism and the mechanical behavior of an additively manufactured (AM) carbon fiber-reinforced polyamide (PA6). Different pressure and temperature levels during HTC were tested to determine the overall effect on the mechanical behavior and material crystalline composition. Treated, carbon fiber-reinforced PA6 samples were analyzed using differential scanning calorimetry, X-ray diffraction, thermogravimetric analysis, scanning electron microscopy, and three-point bending testing. When considered with respect to as-printed samples, an HTC temperature of 190 °C combined with 80 psi pressure resulted in an increased flexural modulus and strength of 47% and 58%, respectively. This increase was attributed to the decrease in AM-induced cracking, voids (both inside and between the beads), and crystalline solid-state transition in the PA6. The effect of pressure and temperature on the crystalline structure was discussed in terms of an increased degree of crystallinity and the amount of α-phase. Therefore, HTC can help overcome some limitations of traditional annealing, which can result in recrystallization-induced cracking which can lead to material embrittlement. The proposed HTC method demonstrates the potential in improving the mechanical behavior of AM thermoplastic composites.

## 1. Introduction

Additive manufacturing (AM) excels as a cost-effective method for prototyping and fabrication. Fused deposition modeling (FDM) is a type of AM process where successive layers of material are deposited upon previously deposited layers until a part form is completed [1]. FDM of fiber-reinforced thermoplastics allows for the custom deposition of materials during the printing process [2,3]. The build material is heated and extruded through a nozzle and the deposited material is bonded to the previous layer via a fusion process. A common and recognized issue with the AM FDM process is the presence of voids and the restricted degree of fusion bonding between the deposited layers. FDM-generated parts typically have lower mechanical properties because of the layer-by-layer nature of the process: extruded molten beads come into contact with the previously deposited layer, but the relatively rapid cooling prevents complete fusion between the beads, leaving voids inside the beads and between their interfaces [1,2,4].

Carbon fibers are commonly introduced into the filament to improve the material strength using short discontinuous carbon fibers (SCF) or long discontinuous carbon fibers (LCF) [2,5,6,7]. As a result, fiber alignment during printing and strength in FDM composites exhibit anisotropy [8]; this is also in part due to incomplete fusion bonding between the printed layers [4,9]. Furthermore, the presence of fiber reinforcement, which increases the material viscosity, was found to result in higher void content both within the layers of filament (intra-bead voids) and between beads (inter-bead voids) [2,10,11]. FDM manufacturing parameters can also affect the formation of defects. Specifically, the major parameters affecting defects are print speed, print direction, layer height, extruder temperature, and print bed temperature [1,12,13,14,15,16]. Higher printing speeds were found to produce parts with greater void content and worsened interlayer bonding [17]. While part anisotropy can be reduced by incorporating quasi-isotropic printing orientations [14,18,19], the presence of FDM manufacturing defects restricts the achievable levels of mechanical performance.

The thermal history and the presence of thermal gradients that occur during FDM fabrication also affect the mechanical properties through crystalline composition [20], also known as polymorphism, as well as affecting the layer fusion [21,22]. Therefore, the fusion bonding during the FDM process is driven by a non-isothermal process, wherein polymer chain reptation across the layer interfaces determines the fusion bond strength [3,23,24]. In this respect, the presence of residual stresses as a result of the FDM process contributes to inconsistent and incomplete fusion bonding along with the crystallization process.

Post printing, high-temperature treatments, such as annealing, have been recently shown to improve the strength and the modulus of elasticity by increasing the interfacial bonding between the deposited beads [9,25,26], as well as the degree of crystallinity [27,28,29,30]. Furthermore, the improvement of interfacial adhesion between filaments [9,30] and the release of locked-in residual thermal stresses through viscoelastic relaxation [30,31,32] can be achieved when annealing AM polymers above the glass transition temperature [9,33]. Other authors considered annealing below the glass transition temperature to avoid recrystallization, which can cause embrittlement in semicrystalline polymers. However, annealing cannot completely reduce the manufacturing defects such as voids (both inter- and intra-bead), poor fusion bonding, and microscopic cracking. In order to remove or reduce these defects, additional pressure is necessary, which will allow polymer chain reptation. High-temperature compaction (HTC), also known as hot pressing, has been studied to determine its effect on the mechanical properties of AM parts [3,20]. HTC is a post-processing method, which is different from in situ compaction using rollers during 3D printing, which typically requires a more complicated printing setup. HTC can be considered as an efficient way to improve the mechanical behavior of AM composites, during which pressure is applied to the already-3D-printed part at elevated temperature, thus reducing the void content [3]. In prior work, the HTC of polyamide (PA6) composite showed significant reduction in void content and increased crystallinity, leading to improvements in flexural strength, while resulting in reduced fracture toughness [3,34]. The reduction in fracture toughness was attributed to an increased level of crystallinity [34].

Therefore, HTC parameters, such as pressure and temperature, affect not only the morphology of the FDM composite, but also its crystalline content, both of which have significant effects upon mechanical properties [35,36]. While prior studies focused on the changes in crystallinity due to thermal gradients during FDM [32], and previously discussed annealing, limited knowledge exists on the combined effects of pressure and temperature during HTC upon polymorphs within FDM composites. Specific to PA6, both the α-type and the γ-type crystalline phases coexist [35], each with its distinct mechanical properties [37,38]. Therefore, to assess the role of crystallinity and different crystalline structures on the resulting mechanical behavior requires capturing the associated phase transition in FDM composites during hot treatments.

The present study investigates the effects of HTC on PA6 carbon fiber-reinforced composites produced by the FDM process. The aim of this work is to study the effect of HTC upon the polymorphism and the mechanical properties of an AM carbon fiber-reinforced PA6. The effects of HTC upon the tensile strength and the elastic modulus of an AM carbon fiber-reinforced PA6 has previously been explored [3]. This work builds upon that foundation by exploring how HTC affects flexural properties and void closure, and then explains those changes in material properties by showing changes in crystallinity content and structure. The effect of pressure and temperature during HTC was explored in terms of the dimensional stability, mechanical properties, and PA6 crystalline polymorph compositions. Since crystallization-induced shrinkage can result in reduced mechanical properties, especially at elevated annealing temperatures [3,20], the crystalline structure of PA6 was quantified using the results of differential scanning calorimetry (DSC) and X-ray diffraction (XRD). Microstructural analysis was used to evaluate the effects of HTC on the microstructural features and defects. These characterization techniques allowed for the demonstration of the role of HTC parameters (pressure and temperature) upon the material and crystalline morphology, as well as its connection to the mechanical properties. The present study demonstrates that HTC can be effective to significantly improve the mechanical properties of AM composites, while controlling the dimensional stability of the FDM part.

## 2. Experimental Methodology

### 2.1. Flexural Test Sample Configuration

Flexural test samples were printed using a combination of PA6 with SCF and PA6 with LCF utilizing the Mark Two FDM printer manufactured by the Markforged Holding Corporation (Waltham, MA, USA). The Markforged Mark Two is a dual extruder printer allowing the incorporation of SCF and LCF layers into the print. The Mark Two uses a minimum of two layers of SCF to separate layers of LCF to achieve a higher quality of bonding between the deposited layers [35,36,39]. The SCF PA6, with trade name Onyx, was purchased from Markforged [5]. The composition of Onyx was previously reported in Ref. [40], including fiber diameter and orientation. LCF filament has a predominantly unidirectional [40] fiber orientation, while the fiber orientation within the SCF filament has more of a 3D orientation with some degree of alignment in the printing direction. A summary of material details for both SCF and LCF filaments are shown in Table 1. It is noteworthy that LCF filament uses higher carbon fiber content (45% by volume) compared to SCF filament (10.5% by volume). The LCF was deposited using a heated extruder set to 270 °C with a print head speed of 15 mm/s, and the SCF was deposited at 265 °C with a print head speed of 20 mm/s. Each printed layer of each of the test samples was set to be 0.125 mm high within the slicer software.

A panel shape with dimensions of 101.6 mm (4 inches) × 101.6 mm (4 inches) × 3.75 mm was generated. The stl file generated using Creo software was exported to the slicer Eiger, which is used by Markforged. All panels were fabricated with the following LCF reinforcement configurations: a quasi-isotropic LCF orientation, a transverse LCF orientation, or a longitudinal LCF orientation. For the quasi-isotropic configuration, the LCF layer angles varied throughout the thickness following a [0°/45°/−45°/90°] orientation. This was selected to ensure effective isotropic behavior, thus reducing the reinforcement directional dependence of the material properties. The unidirectional prints consisted of a 0° or 90° LCF orientation for longitudinal or transverse samples, respectively. The quasi-isotropic configuration is referred to as SCF/LCF-QI and both the transverse and longitudinal unidirectional are referred to as SCF/LCF-UD. After panel fabrication and HTC treatment, the panels were cut into four samples for 3-point bending tests with the approximate dimensions of 25 mm (1 inch) × 101.6 mm (4 inches) × 3.75 mm. A waterjet OMAX ProtoMAX was used to cut the samples to the desired dimensions. Before testing, the sample edges were coated with a white spray paint, followed by drying in the Carbolite GERO industrial oven for twenty hours at 40 °C. A side view of the flexural sample used in this study is provided in Figure 1a, and the cross-sectional view, showing the individual layer configuration, is shown in Figure 1b.

The SCF/LCF-QI samples were used to test the effects of compaction pressure and temperature on the overall mechanical behavior of the samples. This was accomplished by varying the levels of pressure—15 psi (ambient atmospheric pressure), 80 psi, and 100 psi— and temperature—155 °C, 175 °C, 190 °C, and 200 °C. Based on the results of the QI sample mechanical testing, an HTC pressure of 80 psi was selected and was utilized at the above-stated temperatures to determine the mechanical properties in the principal directions by controlling the UD orientation of the LCF filament.

### 2.2. Fabrication of 3D-Printed Single-Layer Films for Material Characterization

Single 3D-printed layers were used to study the effect of HTC and annealing on the crystalline structure of PA6, as well as the resulting crystallization-induced shrinkages. The fabricated single layers of LCF and SCF had initial measured thicknesses of 192 μm and 183 μm, respectively. The treatment conditions, pressure and temperature, of the single-layer films were identical to the treatment of the fabricated panels that were used for flexural testing. Figure 1c,d show a single layer of SCF and LCF sample prior to treatment. The use of single-layer films allows for the observation of any cracking due to recrystallization-induced shrinkage that may develop during HTC or annealing, as will be discussed later.

### 2.3. HTC and Annealing Process

The single-layer films and the fabricated panels were both treated using a pneumatic hot press (Bubble Magic Rosin) for a period of 4 h. The samples were placed directly between the platens of the hot press. Annealed samples were treated inside the convective oven (Carbolite GERO, Parsons Lane, Hope, UK) for 4 h. Once again, the selected temperatures for HTC and annealing were 155 °C, 175 °C, and 190 °C, covering the zones below, near, and within the melting region, since the melting of this grade of PA6 SCF filament was found to start at around 180 °C and be complete at around 210 °C. After the treatment, the samples were taken out from the press or oven, allowing them to cool down to the ambient air temperature. Since the samples never experienced full melting during annealing or HTC, they retained their shape. The 200 °C temperature was eliminated due to excessive sample deformation, which will be discussed later.

### 2.4. Thermogravimetric Analysis

Thermogravimetric analysis (TGA) was employed to investigate any potential thermal degradation in the SCF material resulting from the HTC and the annealing process. During the TGA process, the sample was heated while continually monitoring the change in weight. TGA also allowed the measurement of the overall fiber weight fraction. A small piece of pristine or treated SCF film was placed in a controlled nitrogen atmosphere with a flow rate of 50 mL/min [41]. The TGA was performed using a TA Instruments Q5000 IP TGA analyzer (New Castle, DE, USA) with a heat rate of 10 °C/min. Weight loss over a temperature range from 32 °C to 714 °C was used to analyze the material decomposition. The first derivative of the TGA weight loss curve with respect to temperature (DTGA) was used to identify the peak of the mass rate degradation.

### 2.5. Differential Scanning Calorimetry Analysis

Differential scanning calorimetry (DSC) measured the heat flow needed to change the temperature in pristine and treated single layers of SCF film. In order to understand the effect of annealing and compaction on the changes in the crystalline composition of AM composites, DSC analysis was performed on single SCF layers after they were printed and treated. The DSC technique was employed to determine the first- and second-order transition temperatures, including melting and glass transition temperatures, while also capturing the exothermic and endothermic behaviors during cold crystallization and melting [38,42]. The samples were subjected to a nitrogen flow of 50 mL/min with a heating rate of 10 °C/min, covering a temperature range from −60 °C to 250 °C. The heat required to change the temperature of each prepared SCF was measured against a standard reference material. The crystallinity fraction ($\chi_{c}$) of the specimen was determined using Equation (1) based on the heating cycle.(1)$$\chi_{c}=\frac{\Delta H_{m}-\Delta H_{C}}{C_{m}\Delta H_{m}^{100}}\;x\;100$$
where $\Delta H_{m}$ and $\Delta H_{C}$ are the crystalline melt enthalpy and the cold crystallization enthalpy obtained from the DSC heat scan. $\Delta H_{m}^{100}$ and $C_{m}$ are the enthalpy of melting of fully crystallized PA6 (corresponding to 135 J/g [38,40]) and the mass fraction of PA6 in SCF, respectively.

### 2.6. X-Ray Diffraction Analysis

X-ray diffraction (XRD) analysis was utilized to examine the crystallographic structure and phase composition of PA6. The XRD measurements were conducted by exposing the samples to X-ray radiation with a wavelength (λ) of 1.5418 Å, which interacts with the atomic structure, generating characteristic diffraction patterns. These patterns provide insights into atomic spacing, crystallographic phases, and the degree of crystallinity in the material. All samples were mounted on a zero-diffraction background plate and scanned within a 2-theta range of 6 to 76 degrees. The XRD data were processed using OriginPro software, as PA6 exhibits a semi-crystalline nature where amorphous and crystalline phases overlap. To differentiate these phases, an asymmetric least squares smoothing baseline was applied, using a smoothing factor of 3 and a threshold of 0.05. A coefficient of determination (R^2^) of approximately 97.8% was maintained to ensure an accurate data fit. These baseline parameters remained consistent across all annealed and HTC samples. Equation (2) is used to calculate the degree of crystallinity:(2)$$\chi_{X}=\frac{A_{C}}{A_{A}+A_{C}}$$
where *A_C_* and *A_A_* are the area of the crystalline peaks and the area of the broad amorphous peak (baseline of the crystalline peaks). Further analysis of the XRD data was conducted to evaluate the impact of annealing and HTC on microcrystal lamella changes. The mean size of the microcrystal lamella in PA6 was determined using the Scherrer equation [41], as presented in Equation (3).(3)$$L=\frac{K\lambda }{\alpha_{f}\;cos(\alpha )}$$
where *K* is a dimensionless shape factor representing the normalized variation of the actual shape of the crystallite. *α_f_* and *α* are the full width half maximum intensity (FWHM) and Bragg angle, respectively.

PA6 typically exhibits two main crystal structures [43], namely the α-type and γ-type polymorphs, with the monoclinic α-crystalline phase being the most thermodynamically stable, while the γ-phase adopts a less stable pseudohexagonal structure. The XRD analytical technique allows the characterization of the types of crystals by representing each unit cell with distinct Miller indices for its axes. Miller indices are used in crystallography to describe the orientation of crystal planes and the direction in a crystal lattice. The notation “*h*”, “*k*”, and “*ℓ*” represents a set of three integers that define a specific (parallel) lattice plane in a crystal for a crystallographic axis. The micro-structure hierarchy, along with the α-type PA6 unit cell, is depicted in Figure 2. The unit cell’s characterization entails three axes denoted as *a*, *b*, and *c*. The *a*-axis represents the plane of hydrogen bonding, enabling interactions between polymer chains, while the b-axis aligns with the molecular chain direction. The c-axis corresponds to the progressive shear axis, signifying hydrogen sheet bonding. The atomic spacing between the hydrogen sheet and polymer chains is denoted as d_a_ and d_c_, corresponding to the *a*-axis and *c*-axis of the monoclinic structure [41], with a shear angle (β). Based on the literature [41,43], the shear angle for the monoclinic structure was estimated at approximately 65°. At the micro-scale, spherulites generate circular grain boundaries, with their branches composed of lamellae containing microcrystal lamellae within the lamella layers. The dimensions of these microcrystal lamellae are calculated using Equation (5). Additionally, the atomic spacing and unit cell dimensions of the monoclinic (α-PA6) crystal were determined using Equations (4) and (5) [41].(4)$$d=\frac{\lambda }{2\;sin\alpha_{1}}$$(5)$$\frac{1}{d^{2}}=\frac{h^{2}}{a^{2}\;{sin}^{2}\beta }+\frac{k^{2}}{b^{2}}+\frac{l^{2}}{c^{2}\;{sin}^{2}\beta }-\frac{2\;h\;l\;cos\beta }{a\;c\;{sin}^{2}\beta }$$

### 2.7. Scanning Electron Microscopy Analysis

An SEM analysis was performed in order to evaluate the void volume fraction in various regions of the composite. The calculation of the void fraction was accomplished by using ImageJ software ImageJ 1.53k. The samples for the SEM analysis were cut from SCF/LCF-QI samples and polished. The images were converted into binary format, with voids depicted as black areas and all other regions as white areas. ImageJ identified each pixel within the binary SEM image as either a black or a white region. ImageJ then calculated the number of black and white pixels, then proceeded to calculate percentages based upon the calculated totals.

### 2.8. Flexural Testing

Flexural testing was conducted via a 3-point bending test using a Tinius Olsen 10ST based upon the procedure described in ASTM D790−15 [44]. The crosshead speed was 4 mm/min, which falls within the range for quasi-static testing. The roller fixture had a 5 mm radius. The 3-point bending span was 60 mm. A total of 5 samples for each treatment condition was tested. The load–displacement data were collected and used for stress–strain analysis. The flexural modulus, stress, and strain are calculated using Equations (6)–(8). The flexural properties of pristine, annealed, and HTC composites were compared to determine the effect of high-temperature treatment on the mechanical behavior of the composites.(6)$$E_{f}=\frac{L^{3}m}{4wd^{3}}$$(7)$$\sigma =\frac{3FL}{2wd^{2}}$$(8)$$\varepsilon_{f}=\frac{6Dd}{L^{2}}$$
where *w* is the beam width, *d* is the beam thickness, *D* is deflection in the sample at the roller head, *F* is the force applied, *L* is the span, and *m* is the force vs. position slope near the beginning of the flexural test, during elastic deformation.

## 3. Results

### 3.1. Single-Layer Shrinkage Due to Annealing and Compaction

Post-treatment images of SCF and LCF single layers are shown in Figure 3. The results in terms of treating single-layer films were consistent and were observed in all of the experiments. Specifically, there were no observed distortions induced in the single layer film by the annealing or the HTC processes. The main result of the thermal treatment, either HTC or annealing, was found in the cracking of the film which, as will be discussed, was due to recrystallization. The photograph in Figure 3a shows an SCF layer after being subjected to a temperature of 190 °C.

In Figure 3a, the SCF filament layer under annealing, which contains approximately 10% vol. of short (about 100–200 μm) carbon fibers, did not experience significant shrinkage. Furthermore, the LCF layer that had been subjected to annealing at 190 °C, shown in Figure 3b, exhibited significant cracking. As a point of observation, the LCF that was annealed at 190 °C developed cracking while still in the oven, while the LCF film that was subjected to HTC treatment (Figure 3c) did not develop such cracks. Based upon these results, the observed cracking can be attributed to recrystallization-induced shrinkage and the higher anisotropy in the LCF filament, which contains 45% vol. fiber and longer discontinuous carbon fibers (about 900–1000 μm), compared to the short carbon fibers found in SCF.

The anisotropy in shrinkage is expected between the printing and the transverse to the printing direction because of the preferred fiber alignment in the deposition direction during FDM. Since both LCF and SCF utilize PA6 as the matrix, the apparent difference in film cracking is due to the higher fiber volume fraction found in the LCF filament. The effect of shrinkage is evident by the change in the transverse direction dimensions. The average LCF film width reduced by 11% after annealing at 190 °C (from 44.22 mm in pristine to 39.25 mm). In contrast, the HTC-treated LCF film at 190 °C and 80 psi maintained its initial width of 44.70 mm. Therefore, LCF single layers under HTC pressure showed no cracking, as the anisotropic shrinkage was confined and use of an elevated temperature promoted healing, as shown in Figure 3c.

### 3.2. Compaction Strain Due to Annealing and HTC Treatment

In order to control the dimensional stability during the HTC process, it is important to account for the compaction strain. Therefore, the compaction strain in thickness direction, e, for SCF/LCF-QI and SCF/LCF-UD samples was calculated using Equation (9).(9)$$e=\frac{\Delta d}{d}$$
where Δ*d* is the change in the thickness and *d* is the original thickness of the sample. The compaction strain for SCF/LCF-QI samples are tabulated in Table 2. A summary of the compaction strain for unidirectional printed samples is provided in Figure 4a.

The samples that had a higher thickness after treatment, such as the SCF/LCF-QI annealed at 155 °C, 175 °C, and 200 °C, exhibited delamination that developed during the annealing process. An example of a visibly delaminated case is shown in Figure 4b, which is the sample annealed at 175 °C. Comparing Figure 4b to the previous observations for the thin film (Figure 3b), recrystallization-induced shrinkage under incomplete fusion bonding resulted in cracking between the printed layers of the material. The produced delaminations, especially for 155 °C and 175 °C, resulted in the increased values of sample thickness post annealing, as seen in Table 2. For 190 °C, the visible delamination was not observed; however, the mechanical properties remained low when compared to the 190 °C HTC sample, which will be shown later in the discussion. The reason for less pronounced delamination in the 190 °C-annealed sample can be due to increased autoadhesion between the printed filaments [21], which overcomes the observed recrystallization-induced shrinkage.

As discussed previously, recrystallization-induced shrinkage is inherently anisotropic due to the preferred fiber alignment in the direction of printing, resulting in the path transverse to the printing direction being more dominated by the matrix properties [9,18,21]. As seen in Figure 4b, the absence of external pressure during annealing at sufficiently high temperatures leads to substantial cracking. Therefore, the observed delaminations are due to a combination of (a) shrinkage inside of the LCF layers, (b) the difference between the respective shrinkage rates between the LCF and SCF layers due to differing carbon fiber content and (c) incomplete fusion bonding between the layers. However, with the addition of pressure, the cracks originating from both the FDM process and from recrystallization can be healed. However, when the HTC temperature reached 200 °C, the samples exhibited significant loss of geometry, as seen in the high amount of deformation due to 30% compaction strain.

The right balance of temperature and pressure is required to prevent internal cracking, while ensuring dimensional stability. The compaction strain increased rapidly when approaching 200 °C, indicating that the 175–190 °C temperature range and 80 psi can ensure the right balance. Therefore, to further explore the principal mechanical properties, the two temperature regimes of 175 °C and 190 °C at 80 psi were used to test the SCF/LCF-UD samples in the longitudinal and the transverse to the printing directions.

### 3.3. Thermogravimetric Analysis Results

To understand if any potential thermal degradation appears as a result of annealing and HTC treatment, TGA was performed by comparing the mass loss between treated and untreated samples during continuous heating. The TGA results for SCF single layers are plotted in Figure 5a. Rapid SCF degradation began in both treated and pristine samples at approximately 352 °C and continued to approximately 478 °C with a peak degradation of the derivative weight, *dW*/*dT*, at 449 °C, as evidenced in Figure 5b. Therefore, treated and pristine samples showed similar degradation mass loss behavior. Table 3 summarizes the percent weight loss at two temperatures: at 200 °C and 450 °C. All samples showed the same mass loss of about 2% at 200 °C. Mass loss at 450 °C slightly increased from 47% to 53% for the pristine and the 200 °C-treated samples (both annealed and HTC). The remaining weight at 478 °C of about 20% wt. for all SCF layers represented the carbon fiber weight fraction in the SCF filament.

### 3.4. Differential Scanning Calorimetry Analysis Results

The DSC results in Figure 5c,d depict the melting behavior indicative of the overall degree of crystallinity and the underlying crystalline structures in the pristine SCF thin film, as well as after annealing and HTC treatment. The degree of crystallinity was determined by using Equation (1), considering the 80% mass fraction of PA6 within the samples and a melt enthalpy of PA6 at 100% crystallinity, which corresponds to 135 J/g [40]. The pristine sample exhibited an endotherm with a peak (*T_α_*) at 200.7 °C, which is associated with the dominant α-form crystals [43]. An exothermic peak at a lower temperature was observed in the pristine material, indicating a cold crystallization starting at 100 °C. This is indicative of the restricted crystallization during FDM fabrication, as a result of unfavorable cooling during the FDM process that did not allow it to reach the full achievable crystallization. In order to determine the degree of crystallinity in pristine FDM material, this cold crystallization exotherm must be subtracted from the enthalpy of melting.

Figure 5c shows DSC curves for the different HTC conditions. The sample treated at 155 °C showed two distinct endothermic peaks. The larger peak at the higher temperature (*T_α_*) of 204.4 °C corresponds to α-form crystals in PA6, while a lower-temperature endotherm peak ${(T}_{X}^{1})\;$at 166 °C corresponds to another crystalline structure. The HTC-treated sample at 175 °C displayed a similar pattern with two endothermic peaks. An α-form dominated peak was observed at 200.9 °C, and interestingly, the low-temperature endothermic peak ${(T}_{X}^{2})$, previously observed for 155 °C HTC-treated sample, shifted to 181 °C, which indicates a higher degree of perfection in those crystals. For the HTC-treated sample at 190 °C, the presence of two endothermic peaks remains consistent. However, a noteworthy change is that the low-temperature endothermic peak, previously observed in the 155 °C and 175 °C compacted samples, shifts towards the temperature of 191.7 °C, which is representative of the melting of γ-form crystals in PA6 (*T_γ_*). This shift in the lower-temperature endothermic peaks aligns with observations made in [43], where a similar phenomenon was noted during annealing of PA6 at 195 °C for varying durations. The cold crystallization is still present for all HTC conditions, even though not as pronounced as in the pristine SCF material, but the corresponding exotherm is reducing with increasing HTC temperature. This is indicative that the locked amorphous regions that were not able to crystallize during FDM have converted into crystalline phases. Furthermore, the shift in the lower peak temperature towards *T_γ_* points towards an increased degree of perfection in these crystalline phases, something that will be discussed when analyzing XRD results.

Figure 5d demonstrates DSC curves for the different annealing temperature conditions. The material annealed at 155 °C exhibited two distinct endothermic peaks. The higher peak corresponds to the presence of α-type PA6 crystals, while the lower-temperature endothermic peak ${(T}_{X}^{3})\;$occurs at 171 °C. For the samples annealed at 175 °C, they also display a dual peak pattern, although with a domination of α-type PA6. It is noteworthy that the lower-temperature endothermic peak shifted to coincide with the γ-form peak as the annealing temperature increased. This phenomenon was previously observed in [43]. In contrast to the other treated samples, the annealed sample at 190 °C exhibited a singular peak dominated by γ-type PA6 crystals. All the annealed samples displayed signs of cold crystallization without a clear trend in relation to the temperature; compared to pristine, the cold crystallization enthalpy decreased at 155 °C, and increased at higher temperatures (175 °C and 190 °C). Conversely, for HTC samples, the cold crystallization enthalpy decreased significantly and consistently with increasing temperature. The summarized enthalpy of melting and enthalpy of cold crystallization for pristine, annealed, and HTC-treated samples appear in Table 4. Taken together, the DSC results indicate that the differences in the degree of perfection of the crystalline structures between the annealed samples and the HTC samples are due to the effects of pressure. Additionally, the higher temperature of HTC treatment leads to a higher degree of perfection, as seen in the reduced H_c_ with higher temperature, and the increased overall degree of crystallinity.

These results for the thin films were be used in conjunction with the XRD characterization to explain the mechanical behavior of PA6 composites. It is important to note that although there are differences in thickness between thin-film and the flexural samples, which cause differences in the transient heat transfer at the start of thermal treatment, the overall thermal history is nearly, if not exactly, the same due to the prolonged, 4 h, isothermal treatment for both types of the samples. Furthermore, any differences in cooling upon extraction of the samples from the press or oven are unlikely to play a significant role in explaining the experimentally observed polymorphism behavior, since the samples were not molten during the treatment. Therefore, the crystalline transformations were diffusion-controlled and developed over the extended treatment time of 4 h.

### 3.5. XRD Analysis

As previously mentioned, PA6 exhibits polymorphism which is indicative of its thermal processing history. Specifically, for the FDM process, due to unfavorable cooling during layer-by-layer deposition, crystallization of PA6 can lead to incomplete ordered crystal structure formation and restricted crystal growth. The latter was seen in cold crystallization behavior discussed in the DSC results. The XRD analysis allowed for further investigation of the makeup of the polymorph structures in pristine, annealed, and HTC materials. The XRD plot in Figure 6 demonstrates different annealed temperatures with three distinct peaks: α_1_-PA6 (002), γ_1_-PA6 (020), and α_2_-PA6 (200). From these results of HTC in Figure 6a, a clear trend can be observed showing α-type PA6 (α_1_-PA6: 23.75° 2θ and α_2_-PA6: 19.89° 2θ) peaks increasing in height, which is indicative of α-PA6 crystal growth, with increasing temperature. In contrast, the peak for γ-type PA6 (γ_1_-PA6: 21.3° 2θ) was reduced, indicating the transition of γ-PA6 to α-PA6 with an increase in HTC treatment temperature [45,46].

Comparing 155 °C annealed with HTC samples, the α-type PA6 remained largely unchanged, with only minimal increases observed in the γ-type PA6 (as shown in Figure 6b). As the treatment temperatures increased to 175 °C, a reduction in γ-type PA6 was found for the 175 °C-annealed sample, when compared to the 175 °C HTC-treated sample with unchanged α-type (as shown in Figure 6c).

In the case of the 190 °C-annealed and the 190 °C HTC-treated materials, no γ-type PA6 was detected, and the samples were completely dominated by α-type PA6 (as shown in Figure 6d). The initial presence of γ-type in pristine material was a direct consequence of restricted crystallization due to thermal gradients experienced during FDM, which can also be seen as a result of manufacturing-induced residual stresses, while the thermal treatment (both annealing and HTC) led to a solid-state transition from γ-to α-type crystals.

### 3.6. Discussion About Crystalline Polymorph Structure and Degree of Crystallinity

The XRD data were used to determine various parameters, such as the PA6 unit cell dimensions and the size of microcrystal lamella and to correlate them with the DSC results. The atomic spacing (*d_a_* was 4.42 ± 0.01 Å and *d_c_* was 3.74 ± 0.01 Å), derived from Equation (3), remained consistent for all treated samples. The magnitudes of PA6 unit cell dimension “*a*” and “*c*” also remained the same for all treated samples with values of 9.76 ± 0.01 Å and 8.39 ± 0.01 Å, respectively. These results were consistent with the findings reported in [37,38,47]. For the annealed samples, it was observed that the dimension *L_c_* (derived from the α_2_-PA6 peak) experienced a reduction from 68.8 Å to 60.65 Å with an increase in the annealing temperature (as shown in Figure 7a). In contrast, *L_a_* (obtained from the α_1_-PA6 peak), increased with an increase in annealing temperature from 93.88 Å to 124 Å. In general, for annealed samples, an increase in *L_a_* corresponded to a higher α-PA6 crystalline content in SCF, as documented in [41], where α-PA6 crystalline content is the ratio of the α-PA6 crystalline fraction from XRD to the overall crystalline fraction from DSC. A correlation trend is noted here in the annealed samples, wherein the α-PA6 crystalline content rose to 44.2%, 62.7%, and 100% with temperatures of 155 °C, 175 °C, and 190 °C, respectively. For HTC materials, a decrease in *L_c_* and increase in *L_a_* was observed with an increase in temperature (as shown in Figure 7a). Interestingly, the HTC-treated sample also followed the correlation between *L_a_* and α-PA6 crystalline content, leading to an increase in α-PA6 from 47.5% to 59.6% with increasing temperature. Furthermore, the length of microcrystal lamella was greater for HTC conditions compared to the corresponding annealing temperatures.

The crystalline fraction analysis from XRD and DSC data is depicted in Figure 7b–d. As shown in Figure 7b, among the HTC-treated samples, the total crystallinity fraction determined from XRD remained constant, but there was an increase in the fraction of α-type, rising from 18.6% to 23.8%, while the γ-type PA6 fraction decreased from 4.13% to null as the HTC-treated temperature increased. Similarly, the XRD data of annealed material displayed a similar trend, with the α-type PA6 crystallinity fraction increasing from 21.36% to 24.44% and the γ-type PA6 crystallinity fraction decreasing from 4.55% to null as the annealing temperature rose (Figure 7c).

The overall crystallinity fraction measured from the DSC melt enthalpy increases to 39.08% for the 190 °C HTC-treated sample compared to 21.23% for the pristine sample (as shown in Figure 7d). In the case of annealed samples, the maximum crystallinity fraction of 48.32% was attained at an annealing temperature of 155 °C and reduced to 24.3% for the 190 °C-annealed sample. It is important to point out that this analysis of the degree of crystallinity based on DSC results assumes a constant enthalpy of melting for the different crystalline structures of PA6, which is not necessarily an accurate assumption [48]. However, an observed shift in the peak melting temperature with an increase in treatment temperature is indicative of the refinement of the crystalline structure of PA6 and increased degree of perfection.

The insight into this polycrystalline transformation can be seen in the XRD data which show that higher temperatures provide sufficient energy for the γ-type PA6 to transform into α-type PA6, and provide a higher order (degree) of perfection of α-type PA6 for both annealed and HTC-treated samples. Despite HTC samples demonstrating a higher overall crystalline fraction, based on DSC results, the proportion of α-type PA6 remains equivalent to that of annealed samples, based on the XRD analysis. Notably, the HTC treatment provides the material with a higher order of α-type PA6 and longer microcrystal lamella compared to annealed samples.

The discrepancy between the reported degree of crystallinity between DSC and XRD can be attributed to the following. DSC measures the overall crystallinity fraction, which includes all types of crystal forms with varying degrees of perfection, while XRD analyzes the specific crystalline fractions contributed by each crystalline phase. In this study, XRD determined specific crystalline phases, namely the γ-type and α-type PA6. In summary, DSC and XRD analyses demonstrate the agreement that the temperature of 190 °C and time of 4 h were adequate for completing the crystalline transformations.

### 3.7. Scanning Electron Microscopy Results

SEM micrographs of polished cross-sections of SCF/LCF-QI samples are shown in Figure 8a–d. Representative SCF and LCF layers are noted in Figure 8a. Additionally, both interlayer voids and intralayer voids are depicted as black areas. Image processing of the micrographs was used to collect information on the overall interlayer and intralayer void content in the SCF and LCF layers. A summary plot of the calculated void content for the SCF/LCF-QI samples under all test conditions is provided in Figure 8e. HTC demonstrates a significant effect on void closure, as the void content drops from 3.5% to 1.75%. When considering the two types of fiber reinforcement, the void content was found to be slightly higher in SCF compared to LCF. Also, the void content reached a plateau behavior with increasing temperature, which can be attributed to the increased recrystallization and associated shrinkage of the material at higher temperatures. It is likely that this behavior was prevalent with SCF because of the PA6 content being greater than in LCF.

### 3.8. Flexural Testing Results

#### 3.8.1. Flexural Testing of SCF/LCF-QI Samples

The summary of flexural strength and modulus results for SCF/LCF-QI samples are shown in Figure 9a,b. The SCF/LCF-QI samples treated at 200 °C were excluded due to extensive dimensional loss during HTC and the delamination failure of samples during annealing. Overall, the strength of treated samples increased when compared with the pristine configuration by approximately 38%: 118 MPa in pristine vs. 156 MPa for HTC at 155 °C. Additionally, an increase in the modulus was noted, wherein the modulus increased from approximately 3 GPa for pristine to approximately 4 GPa for both annealed and HTC samples. The largest increase was noted for HTC treatment at 190 °C—80 psi, where the modulus increased to 4.4 GPa and the strength increased to 164 MPa. Based on these results, 80 psi was selected along with the temperature range of 175 °C to 190 °C to further investigate the effects of HTC treatment on the mechanical properties in the principal directions: printing and transverse-to-printing directions.

#### 3.8.2. Effect of Pressure and Temperature on the Mechanical Properties in Principal Directions

While the QI sample configuration was used for evaluating the effects of a broader range of HTC parameters upon the mechanical behavior, to fully understand the mechanical behavior requires testing in the principal material directions, which in the case of FDM composites represents the 0° and the 90° UD LCF sample configurations. Representative flexural stress–strain curves for UD and QI LCF samples are shown in Figure 10a–c for three cases, namely pristine, 175 °C—80 psi, and 190 °C—80 psi. For all these cases, SCF/LCF-UD 0° longitudinal samples had the highest flexural strength and the most brittle behavior, while SCF/LCF-UD 90° transverse samples exhibited the greatest ductile failure and lowest flexural strength. As expected, SCF/LCF-QI samples had flexural strength between the SCF/LCF-UD 0° and SCF/LCF-UD 90° samples. SCF/LCF-QI samples also experienced failure in a less brittle fashion than SCF/LCF-UD longitudinal samples, but were less ductile than SCF/LCF-UD transverse samples.

For ease of comparison, representative stress–strain curves for the SCF/LCF-UD samples were grouped by carbon fiber reinforcement angle, as shown in Figure 11. The average flexural strength and standard deviation was 150.75 ± 19.17 MPa at 175 °C—80 psi, while the average flexural modulus was 4.24 ± 0.65 GPa. The highest mechanical properties were found for HTC 190 °C—80 psi: flexural strength for SCF/LCF-UD 0° reached 340 MPa; with SCF/LCF-UD 90° it was 91 MPa, and flexural modulus was 13.93 GPa and 3.43 GPa for SCF/LCF-UD 0° and 90°, respectively.

It can be seen that as the HTC temperature increased to 190 °C, the SCF/LCF-UD 0^°^ samples showed a 24% increase in strength and modulus compared with pristine, while SCF/LCF-UD 90° samples showed an increase of 28% and 40% for flexural strength and modulus, respectively. The corresponding increase in mechanical properties, strength and modulus, with increasing compaction temperature is due to the previously presented increase in α-phase and reduced void content, which led to increased fusion bonding between the layers, as discussed in the next section. The samples compacted at 175 °C experienced a similar increase in void closure and fusion bonding compared to pristine, however not to the extent observed with the 190 °C HTC samples.

#### 3.8.3. Discussion About Failure Modes Post HTC and Their Effect on Mechanical Properties

An examination of the failure progression within a pristine sample and a 190 °C—80 psi sample demonstrates the role of increased fusion layer bonding upon the mechanical behavior (Figure 12). The pristine sample (Figure 12a,b) demonstrates significant delamination between the 3D-printed layers upon flexural failure, which, as evidenced by the microscopy analysis, is related to the presence of the interlayer and intralayer voids. The extensive cracking shown in Figure 12a,b demonstrates that delamination occurred at both types of interfaces, between SCF–SCF, and SCF–LCF layers. However, more interlayer cracking was present at the SCF–LCF interfaces. The number of interlayer cracks is noticeably less in the HTC sample, which indicates the stronger fusion bonding (Figure 12c,d). This would indicate that a stronger interface contributes to improved stress transfer, leading to the higher mechanical properties observed.

The findings within this work, as described earlier, are in line with the findings from previous research concerning the effects of high-temperature annealing for thermoplastic polymers. Some prior studies demonstrated that the mechanical properties of thermoplastics composites can be enhanced by post-processing using high-temperature conditions. For example, enhancement of tensile strength in AM short-fiber composites was reported due to annealing for both a semi-crystalline polymer, PLA, and an amorphous polymer, PETG [9]. Particularly, when annealed at 90 °C for 240 min, the transverse tensile strength of carbon-reinforced PETG increased from 12.4 MPa to 24.4 MPa. Similarly, after annealing at 90 °C for 240 min, the transverse tensile strength of carbon-reinforced PLA increased from 16.0 MPa to 30.8 MPa. While these are significant increases, the overall attainable level of strength is still relatively low; therefore, mechanisms for improving the mechanical properties of more structural AM polymer composites need to be explored. Additionally, annealing alone is not sufficient to rectify all the manufacturing defects associated with AM layer-by-layer fabrication. In our previous work, we investigated the effect of HTC upon the mechanical properties of a carbon fiber-reinforced PA6 AM composite [3]. Therefore, this study investigated the effect of post-processing parameters, such as temperature and pressure, on the mechanical properties under annealing and HTC. Based on the presented results, HTC using 80 psi at 190 °C produced a 25–30% strength increase in the principal directions, while showing a significant reduction in void content, and increasing the α-phase and its degree of crystalline perfection in PA6 [46].

## 4. Conclusions

In order to increase the mechanical properties of FDM composites, post-fabrication thermal treatment such as annealing can be used to improve the mechanical properties. However, as shown with SCF/LCF-QI samples, the higher annealing temperatures of FDM PA6 composites may lead to recrystallization-induced shrinkages that cause cracking inside of the material. Therefore, the addition of pressure using HTC was explored to promote crack healing and void closure. The effect of HTC parameters, such as temperature and pressure, was investigated, and it was demonstrated that a combination of 190 °C and 80 psi was able to increase the flexural strength by 58% and the modulus by 47% while reducing FDM voids to less than 2% (as demonstrated in the SCF/LCF-QI composite).

DSC and XRD analyses were conducted on the HTC and annealed samples to gain an insight into the processing-induced polymorphism within the crystalline structure of PA6. It was found that the conversion of the γ-phase to a mechanically more favorable α-phase occurred as the temperature of treatments increased, as well as inducing solid-state transition of the amorphous regions into crystalline forms. Additionally, the application of pressure in HTC led to a higher degree of perfection in α-type PA6 crystals and longer microcrystal lamella compared to annealed samples.

The study demonstrated that the use of HTC for FDM composites induces various sources of morphological changes occurring on different scales. From the microstructural level, the void content was reduced along with the healing of cracks, while changes in the crystalline structure of PA6 were different compared to the corresponding annealed samples. The latter resulted in the refining of the crystalline phases through conversion of the γ-PA6 into α-PA6 crystals, which have superior mechanical properties. The combination of these effects led to improvements in the material properties of the FDM layers, as well as improved interfacial bonding between the layers, contributing to higher mechanical properties of the composite.

In summary, HTC provides an avenue for enhancing the mechanical behavior of 3D-printed PA6 composites. It should be noted that the temperature for HTC must be sufficiently high to soften the material to allow the applied pressure to close the voids and achieve full fusion bonding. However, excessive temperature and pressure may result in part distortion. The parameters for the HTC process must be selected carefully, so that recrystallization at high temperature does not lead to material embrittlement and cracking from recrystallization shrinkage; conversely, a low temperature of HTC may not be sufficient for crack healing or may even cause additional cracking.

## Figures and Tables

**Figure 1 polymers-17-00922-f001:**
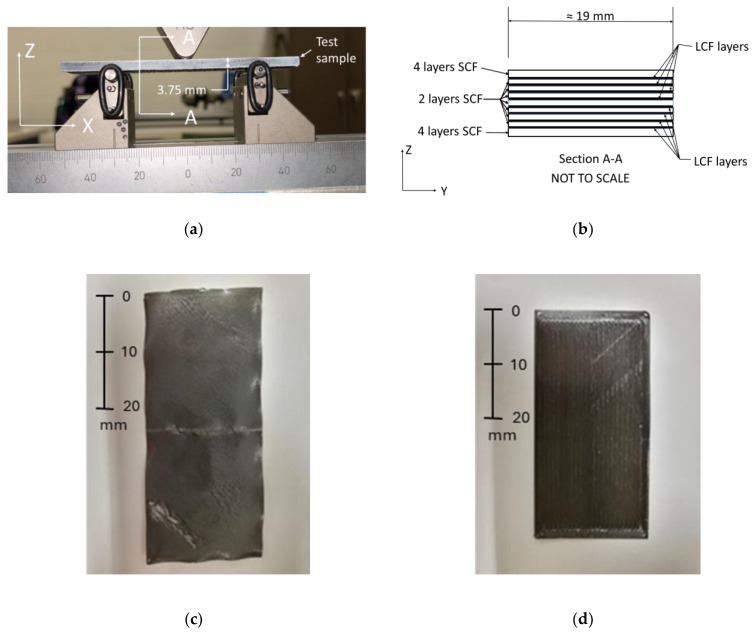
Beam bending sample side view (**a**), individual layer configuration (**b**), SCF single-layer sample (**c**), and LCF single-layer sample (**d**).

**Figure 2 polymers-17-00922-f002:**
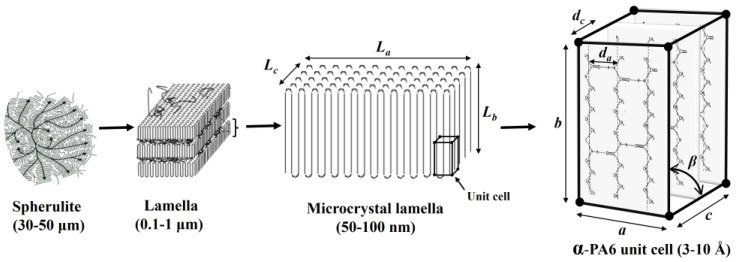
PA6 unit cell and microcrystal lamella.

**Figure 3 polymers-17-00922-f003:**
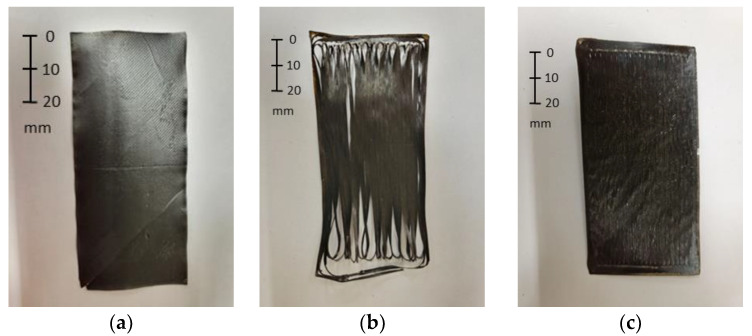
SCF single layer annealed at 190 [°C] (**a**); LCF single layer annealed at 190 [°C] (**b**); and LCF single layer subjected to HTC at 190 [°C] and 80 [psi] (**c**).

**Figure 4 polymers-17-00922-f004:**
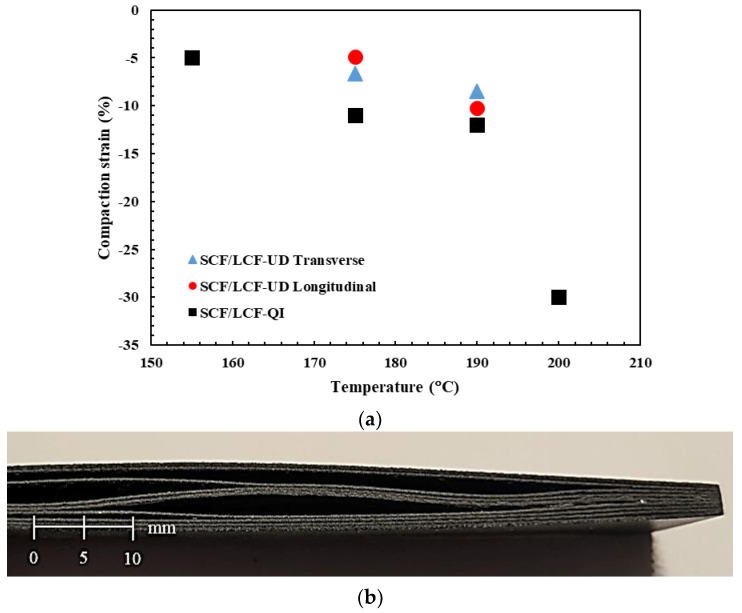
Compaction strain of SCF/LCF treated at 80 [psi] (**a**), and delamination in SCF/LCF sample annealed at 175 [°C] after treatment (**b**).

**Figure 5 polymers-17-00922-f005:**
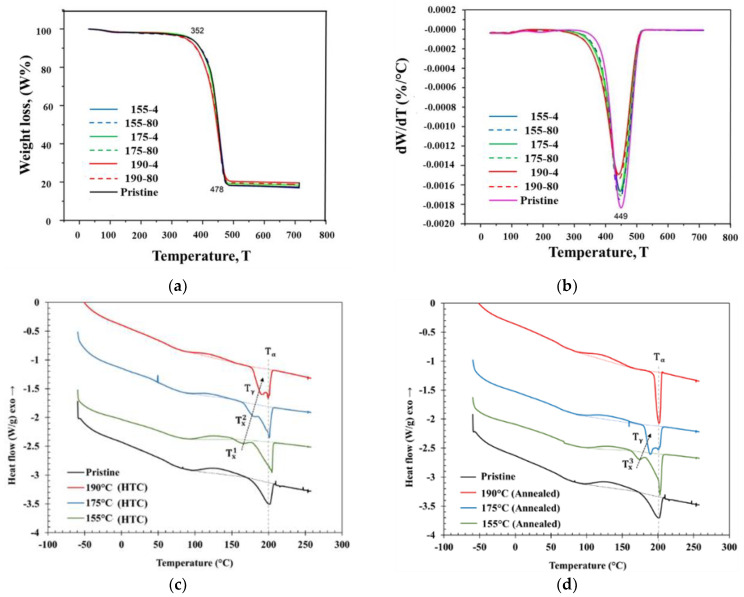
TGA curve for SCF samples treated at various temperatures (**a**), DTGA curves for SCF samples to determine temperature where max weight loss occurs (**b**), and DSC plots for pristine and treated samples (**c**,**d**).

**Figure 6 polymers-17-00922-f006:**
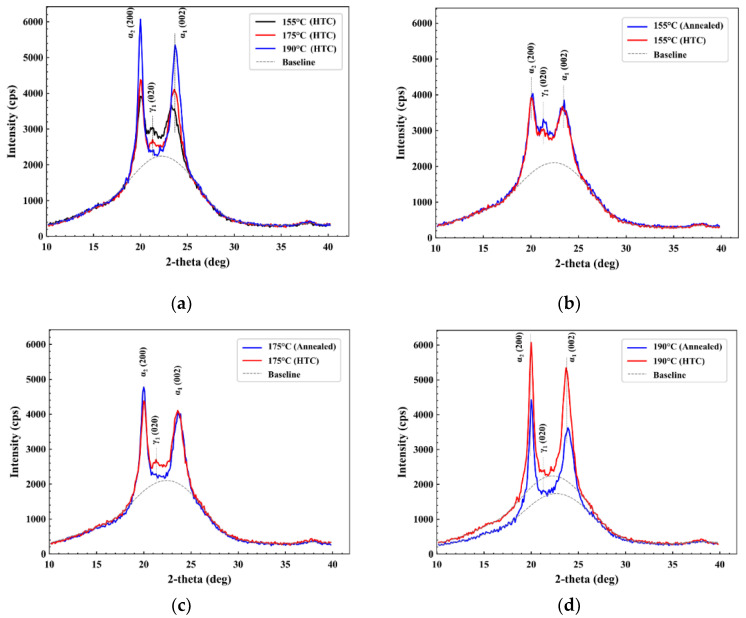
XRD plot for different HTC temperatures (**a**), XRD plot for 155 [°C] annealing and HTC (**b**), XRD plot for 175 [°C] annealing and HTC (**c**), and XRD plot for 190 [°C] annealing and HTC (**d**).

**Figure 7 polymers-17-00922-f007:**
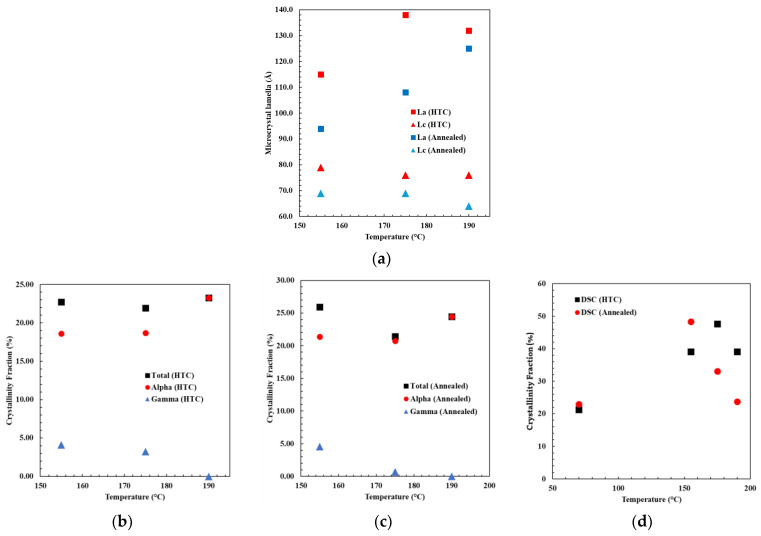
(**a**) α-PA6 microcrystal lamella using XRD data, (**b**) XRD crystallinity fraction of HTC-treated sample, (**c**) XRD crystallinity fraction of annealed sample, and (**d**) DSC crystallinity fraction of HTC-treated sample and annealed sample.

**Figure 8 polymers-17-00922-f008:**
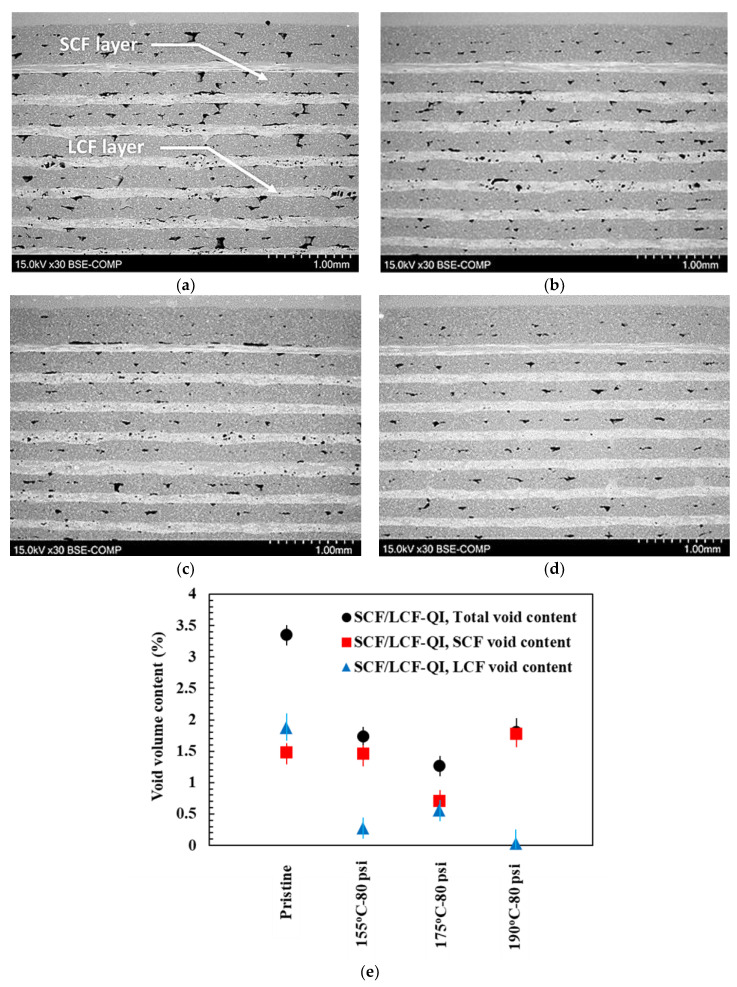
Micrographs of SCF/LCF sample cross-sections for (**a**) pristine, (**b**) 155 [°C], 80 [psi], (**c**) 175 [°C], 80 [psi], (**d**) 190 [°C], 80 [psi], and (**e**) void volume content contribution in SCF/LCF-QI samples.

**Figure 9 polymers-17-00922-f009:**
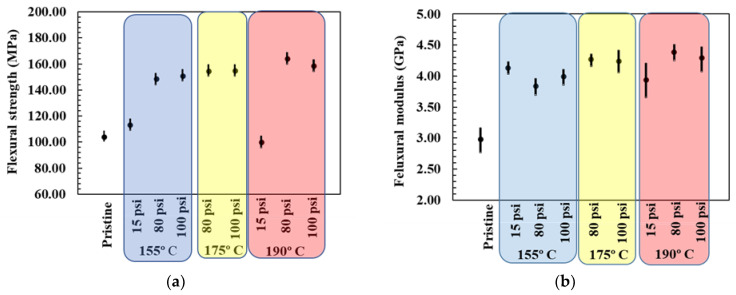
(**a**) SCF/LCF-QI flexural strength and (**b**) flexural modulus.

**Figure 10 polymers-17-00922-f010:**
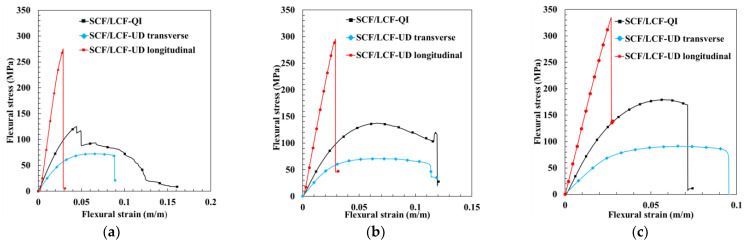
Untreated sample flexural stress–strain curves (**a**), flexural stress-strain curves for samples treated at 80 [psi] and 175 [°C] (**b**), and flexural stress-strain curves for samples treated at 80 [psi] and 190 [°C] (**c**).

**Figure 11 polymers-17-00922-f011:**
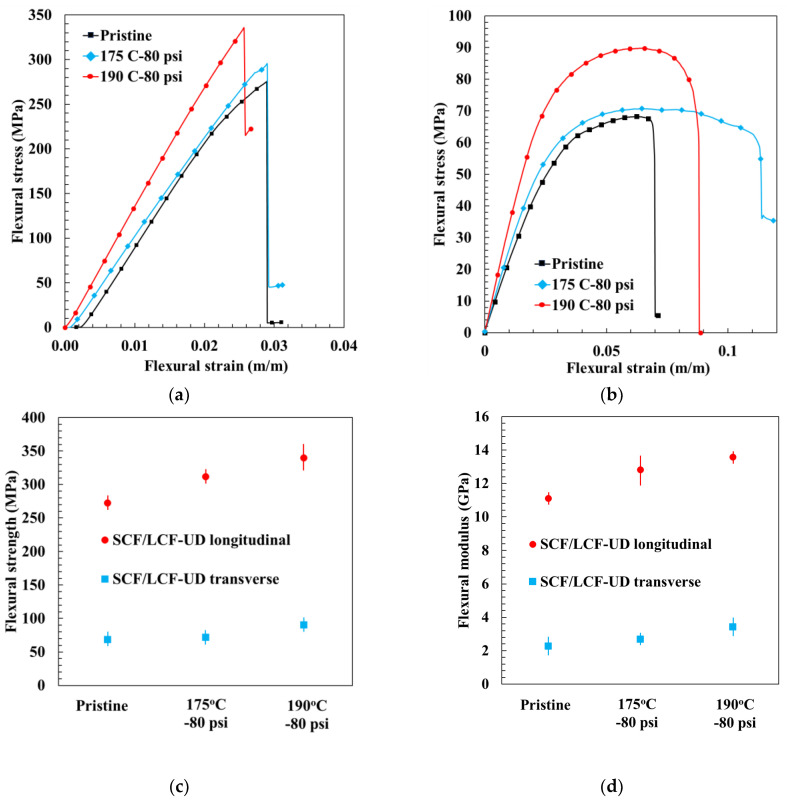
Representative SCF/LCF-UD longitudinal stress–strain curves (**a**,**b**) and SCF/LCF-UD transverse stress–strain curves; summary of flexural strength (**c**) and flexural modulus (**d**) in principal directions.

**Figure 12 polymers-17-00922-f012:**
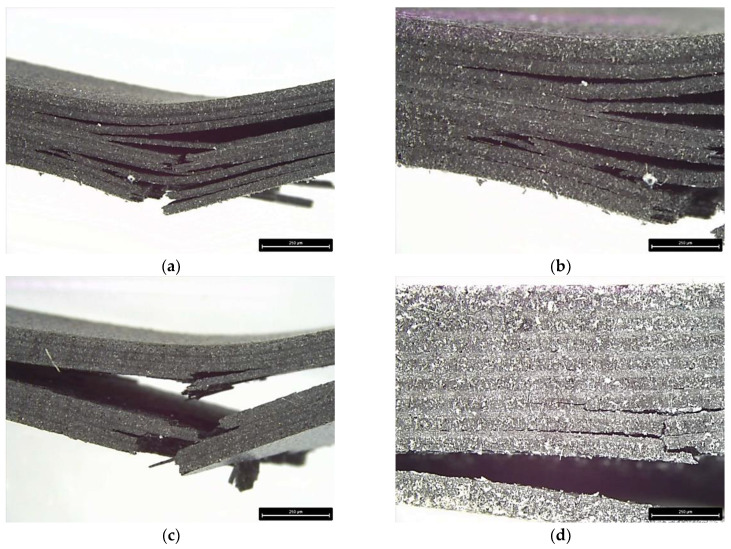
Flexural failure in pristine sample (**a**,**b**) and 190 °C—80 psi sample (**c**,**d**).

**Table 1 polymers-17-00922-t001:** Parameters of short and long carbon fiber filament [40].

Filament Type	Density [g/cm^3^]	Carbon Fiber Vol. [%]	Filament Diameter [μm]	Carbon Fiber Length [μm]
SCF	1.2	10.5	1750 ± 5	168 ± 37
LCF	1.4	45.0	393 ± 2	998 ± 2

**Table 2 polymers-17-00922-t002:** Compaction strain of SCF/LCF samples.

Temperature [°C]	155	155	155	175	175	175	190	190	190	200	200	200
Pressure [psi]	15	80	100	15	80	100	15	80	100	15	80	100
Strain in SCF/LCF-QI	+9%	−5%	−8%	+39% **	−11%	+9%	−3%	−12%	−12%	+48% **	−30% *	−28% *

* indicates sample melted during processing; ** indicates sample visibly delaminated during processing.

**Table 3 polymers-17-00922-t003:** SCF weight loss at 200 [°C] and 450 [°C].

Treatment	Weight Loss (%)	Remaining Weight (%)	Weight Loss (%)	Remaining Weight (%)
	At 200 [°C]		At 450 [°C]	
Pristine	2.08	97.92	47.24	52.76
155 [°C]	2.03	97.97	46.84	53.16
155 [°C]—80 [psi]	2.32	97.68	49.30	50.70
175 [°C]	1.74	98.26	49.90	50.10
175 [°C]—80 [psi]	2.08	97.92	47.24	52.76
190 [°C]	1.85	98.15	53.78	46.22
190 [°C]—80 [psi]	2.08	97.92	52.90	47.10

**Table 4 polymers-17-00922-t004:** Summary of enthalpy of melting for pristine and treated samples.

Sample	HTC at 80 [psi] [J/g]			Annealed [J/g]		
	Enthalpy of Melting [H_m_]	Enthalpy of Cold Crystallization [H_c_]	H_m_ − H_c_	Enthalpy of Melting [H_m_]	Enthalpy of Cold Crystallization [H_c_]	H_m_ − H_c_
Pristine	38.44	15.52	22.93	39.81	15.03	24.78
155 [°C]	52.84	10.61	42.24	61.19	9.00	52.19
175 [°C]	58.52	7.10	51.43	49.62	13.92	35.70
190 [°C]	50.78	8.57	42.21	37.37	12.94	24.43

## Data Availability

The original contributions presented in this study are included in the article. Further inquiries can be directed to the corresponding author.

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
