# Peer review of "Polymorphism and Mechanical Behavior in Hot-Pressed 3D-Printed Polyamide Composite: Effects of Pressure and Temperature"

_polymers, 2025, doi:10.3390/polym17070922_

Round 1
Reviewer 1 Report
Comments and Suggestions for Authors
The manuscript concerns studies on the effect of the hot temperature compaction process on the polymorphism and the selected mechanical properties of carbon fiber-reinforced polyamide PA6. The authors presented the results of studies performed for samples produced at different pressures and temperatures of the compaction process using a wide range of testing methods.
The text is, in general, well organized into chapters. The “Introduction” chapter provides the general background of the problem being studied. The authors presented numerous references to the literature for presenting the state of knowledge on the issues discussed. Unfortunately, the thesis and aim of the studies are not clearly outlined at the end of the chapter, where they typically should be stated. The aim of the work is partially given in the abstract, but this is not the appropriate place for it. The “Experimental methodology” chapter presents raw materials, samples preparation procedures and describes in detail testing methods. The "Results" chapter is divided into subchapters that present the research results from individual methods. Some of the subsections contain a discussion of the results. The manuscript ends with conclusions based on the results and discussion.
The subject of the presented research is interesting, and the results provide curious information about the processing of the tested composite. The selection of research methods, the way to perform research, and the analysis of results are beyond doubt. The results are presented clearly, and their analysis is correct. Unfortunately, the form of presentation of the results and the multitude of editorial errors significantly reduce the readability and quality of the manuscript. The mistakes and errors revealed, and suggestions for improvement in the quality and clarity of the manuscript are presented below:
- As mentioned above, the last paragraph of the "Introduction" chapter should contain the work's precise thesis and aim. Instead, a kind of summary of the research conducted by the authors is presented - this is not the appropriate chapter for such information. Please re-organize the final part of the chapter; in particular, add the thesis and aim of the performed research.
- Subchapter 2.1.: The text is improperly formatted, and the fact that the entire paragraph is bolded is particularly striking.
- A multi-repeated error, "Error! Reference source not found," is throughout the text, making the manuscript's analysis very difficult. Please correct it.
- The Celsius degree symbol is in an unknown format; please use the appropriate symbol U+00B0 (this applies throughout the manuscript text).
- Units are usually written in square brackets. Please check this issue in the entire manuscript and correct it.
- In Figure 1a, the sample thickness marking is illegible. Please consider moving these markings to another place in the drawing - with a uniform, light background.
- The tested sample is not clearly indicated in Fig. 1a; the drawing contains many different elements, and therefore, the indication of the sample is justified.
- The numbering of figures and tables and their references in the text are incorrect; for example, Figure 1 or Table 1 are repeated multiple times. Please review the entire manuscript in this regard and make corrections.
Additionally, some typos corrections are needed to increase the manuscript's quality, for example:
- abstract - …“and the amount of a-phase.”… : Shouldn't be the α-phase?
- “Introduction”, second paragraph - ..”the major paraments affecting defects are”… : Shouldn't be “parameters”?
- “Introduction”, last paragraph - …” differential scanning colorimetry (DSC)”… : Should be “calorimetry”
- Table 1 - (g/cm3): no superscript
- Figure 7a : The unit of length of microcrystal lamella is probably Angstrom, not Ampere. Please correct the unit symbol.
- Subsection 3.8.3: Incorrect subsection heading formatting; please correct it.
- There is no information about the author's contributions or Conflicts of Interest.
- Please arrange and correct the literature and format according to the journal's editorial guidelines.
Reviewer 2 Report
Comments and Suggestions for Authors
The study presents scientific interest, and various experimental methods have been conducted. However, significant improvements are needed in the presentation of certain results, as well as in adhering to MDPI's guidelines.
- Please ensure the manuscript follows MDPI’s formatting guidelines, including sections, reference formatting (both in-text and in the reference list), figures, equations, and text styling (e.g., bold formatting). Resolve errors such as "Error! Reference source not found". Major effort is required to improve this section, as inconsistencies make it difficult to follow the paper.
- Carefully double-check the spelling and grammar throughout the manuscript.
- Maintain a consistent naming convention for the fabricated materials across the paper.
- When possible, use the same colors in graphs to represent the same materials, ensuring clarity in data presentation.
- What is the novelty of the paper in comparison with other works? Indicate it along the paper.
- The results are well presented, but the discussion is weak. A more in-depth comparison with related studies is necessary to strengthen the analysis in most of the results. Expand the discussion on the results, integrating comparisons with relevant literature to highlight the significance of the findings.
- Check the naming and units of the axes in the TGA results. The figure is referred to as Figure 1, but there seems to be a formatting issue.
- Clarify the experimental procedure for calculating the void volume content. How many images were analyzed? Provide this information to reinforce the reliability of the data. Figure 8 lacks error bars. Please include them to reflect data variability and measurement accuracy
- Do not include references in the conclusions section, as per MDPI guidelines.
- Carefully double-check the spelling and grammar throughout the manuscript.
- Some result explanations are difficult to understand.
Round 2
Reviewer 1 Report
Comments and Suggestions for Authors
The quality of the manuscript has improved considerably after the corrections. However, a mistake may be related to my imprecise comment: it concerns units and brackets. The change in the bracket type was supposed to be tables and figures, but not the text. In the text, brackets should not be used for units. Please restore the previous form of the main text related to the brackets (units without brackets).
Reviewer 2 Report
Comments and Suggestions for Authors
The authors have taken into account all my comments.
